# Prevalence of Low Back Pain among Nursing Staff in Najran, Saudi Arabia: A Cross-Sectional Study

**DOI:** 10.3390/medsci8040045

**Published:** 2020-10-30

**Authors:** Adel Alshahrani

**Affiliations:** Rehabilitation Sciences Department, Physical Therapy Program, Najran University, Najran 55461, Saudi Arabia; amsalshahrani@nu.edu.sa

**Keywords:** low back pain, epidemiology, nurse, prevalence, risk factors

## Abstract

This study aimed to determine the various demographic and work-related professional characteristics associated with low back pain among nursing professionals in Najran, Saudi Arabia. A self-administered modified questionnaire (electronic), which included information on general subject demographics and work conditions, was sent through various electronic channels to which 187 nurses working across various health institutions in the Najran region responded. Among the included respondents, 35.3% were Saudis, 64.7% were Non-Saudis, and 88.8% were in the young to middle-age group (21–40 years). Similarly, 57.8% were females, while a majority (91%) had completed a bachelor’s degree. In total, 140 respondents (74.8%) reported experiencing low back pain. Among the various work-related factors, gender, place of work, nature of work, and direct contact hours with patients per week were found to be significantly associated with low back pain. Assessment of pain characteristics found that a majority (88.2%) had mild to moderate localized back pain. A significant number of nursing professionals included herein reported to have low back pain, which appeared to be moderated by work-related characteristics, including place and nature of work. Our findings can help to establish policies and interventions aimed at reducing the risk and onset of low back pain.

## 1. Introduction

Low back pain (LBP), also known as low back ache (LBA), is a very common musculoskeletal condition that has garnered worldwide attention [1,2]. LBP has a prevalence of 84% and ranks 6th on the global burden of disease with 40–60% of individuals with acute LBP still reporting pain after a year and 5–7% having chronic pain [3,4]. Moreover, studies have observed that, on average, 45–70% of the adult population suffer from varying degrees of LBP of mechanical origin at least once during their lifetime [5]. LBP adversely affects vast populations by deterring activities of daily living (ADLs) and preventing a person from leading a physically, psychologically, economically, and socially fulfilling life [6]. LBP has been associated with a sedentary lifestyle, irrespective of gender, nationality, and a variety of other associated factors [4,6].

LBP significantly impacts society by reducing quality of life, and through its large socioeconomic influence. For instance, between 1990 and 2015, disability-adjusted life years (DALY) due to LBP increased by 54% worldwide, with a large proportion of the increase having been observed among countries in the Middle East, Africa, and Asia (low- and middle-income nations) who have inadequate resources and expertise in health and social systems needed to address this growing burden [7]. In Australia, the cost associated with LBP in 2001 exceeded AU $1 billion [8], with LBP having a lifetime prevalence of 11–84%. LBP remains one of the most frequent causes of disability worldwide, which claims around 80% of total expense [9].

In a review of 40 studies investigating the prevalence of persistent LBP in Asia, Africa, and South America (*n* = 80,076), Jackson found that chronic LBP was two and a half times more common among individuals with formal employment than those with non-formal employment [10]. Another study showed that lifestyle, psychological, and social factors were related to work-related LBP [10,11].

Studies have apparently shown that some professions are more likely to suffer from LBP given the nature of their work [5,6,11]. One such profession is nursing, which is physically challenging and has been associated with long working hours, prolonged standing, frequent bending, stooping, lifting, and other risk factors [12,13]. In a series of studies, including one conducted in Turkey, Karahan et al. determined that LBP onset affected as many as 77% of nurses [14]. Yet another study by Chiwaridzo et al. conducted in Zimbabwe reported that 67.9% of nurses experienced varying degrees of LBP over 12 months, while 52.7% reported that the initial episodes occurred during the first five years of working as a nurse [15].

Nurses have been the backbone of the entire health care system and remain under intense physical stress, which may consequently promote LBP. Considering the limited studies on LBP among nurses in Saudi Arabia [11,16,17,18], the present study aimed to primarily determine the prevalence of work-related LBP and its association with various work-related factors among nursing professionals in Najran region, Kingdom of Saudi Arabia.

## 2. Methods

This descriptive cross-sectional study utilized a self-administered questionnaire modified from Dionne et al. [19]. The study population consisted of nursing professionals working at hospitals and health care centers within the Najran region. A pilot study was conducted on 10 nurses with no major modifications required. The questionnaire obtained information regarding socioeconomic and demographic profiles of the nursing professionals, irrespective of their nationalities. Questions included herein were aimed at understanding LBP with respect to onset, type, nature, work settings, and impact of the nature of work on LBP.

The questionnaire was distributed electronically through various social media sites from January 2019 until June 2019. 187 nursing professionals working at clinics, hospitals, community care centers, and schools responded. Data from concerned professionals were also collected using various social media networks in order to include responses from multiple health care locations. Socioeconomic and work-related characteristics were then compared according to the prevalence of LBP irrespective of sex. Similarly, this study aimed at comparing vocational and non-vocational LBP according to gender. This study had been approved by Najran University Ethics Committee for Human Investigations, ethical number 01-10-01-2010 EC, and electronic consent was obtained from all participants.

Data were initially coded and keyed into a Microsoft Excel spreadsheet and then transferred to SPSS software for data processing. Descriptive analysis was used to summarize categorical data. The Pearson chi-square test was used as part of inferential statistics to determine the degree of association between categorical variables. A *p* value of <0.05 indicated significant differences between variables.

## 3. Results

Among the 187 respondents, 35.3% were Saudis, while 64.7% were not non-Saudis, with a maximum 88.8% of the sample population being in the young age group (21–40 years) (Table 1).

A total of 108 respondents (57.8%) were female, while approximately 91% of the participants had a diploma or a bachelor’s degree in nursing. Moreover, 58.8% had a work experience of <10 years, whereas only 10.2% had a work experience of more than 19 years. After comparing various sociodemographic and work-related characteristics and determining their association with LBP, certain findings were noted (Table 2).

Among the 187 participants, 140 complained of LBP, whereas 47 did not. Accordingly, more non-Saudi nurses had LBP (67.8%) compared to their Saudi counterparts irrespective of sex, though the difference was not significant (*p* < 0.05). Moreover, the majority of the nurses (76.2%) who developed some type of LBP were between 26 to 40 years old, with only 7.5% and 1.8% of those aged 41–50 and >50 years developing LBP, respectively. However, no significant association was found between age and LBP (Table 2). Furthermore, LBP was significantly more prevalent among female (60.8%) than male (39.2%) nurses (* *p* < 0.05) (Table 2).

Among nursing professionals with LBP, nearly all (92.8%) had a diploma and a bachelor’s degree, whereas only 7.2% had a master’s degree and higher. However, no significant difference in educational level was noted.

Our data showed that place of work, nature of work, and direct contact hours with patient per week, but not work experience, were statistically associated with LBP (Table 2). Overall, among nursing professionals, 80% of those who developed LBP were employed full time, whereas only 20% were employed part time, with the difference being significant (* *p* < 0.05). Interestingly, our results showed that increased contact hours with patients per week was significantly associated with an increased number of nurses with LBP (* *p* < 0.05).

The final part of the study aimed at exploring the characteristics of LBP reported by nurses from Najran region.

A majority of the nurses (89.3%) reported no past medical conditions, whereas 3.6%, 2.9%, and 4.3% reported having cardiac conditions, hypertension, and other associated medical conditions, respectively (Table 3). Moreover, 59.3% and 40.7% of the participants reported having acute and chronic LBP, with 58.4%, 25%, and 18.6% reporting an average pain duration of 1 week, 2–4 weeks, and >4 weeks, respectively.

A majority of the participants (88.2%) reported having mild to moderate LBP, whereas the rest (17.8%) reported severe LBP. Moreover, a majority (66.4%) reported having no functional limitations, while 23.6%, 7.1%, and 2.9% reported having mild, moderate, and severe functional limitation, respectively. Furthermore, a majority (73.6%, *n* = 103) of the participants reported that LBP had an effect on the execution of normal daily activities (Table 3). Our results showed that the majority of the participants (51.4%) suffered pain localized in the lower back, whereas 38.6% reported having pain in both the lower back and buttocks. However, 10% of the nurses reported having radiating LBP, which suggested peripheralization (Table 3). Our data revealed that 27.2% of the nurses experienced difficulty in lifting and carrying weight, 24.2% reported having problems in prolonged standing, and 20.3% reported having problems in handling patients (Table 3).

## 4. Discussion

The present study focused on work-related factors affecting LBP among nursing staff. Among the various health care professionals, nurses experience tremendous physical stress by virtue of their work responsibilities. Najran, an important city located in the Southwestern part of the Kingdom of Saudi Arabia sharing a border with Yemen, has a number of important hospitals and other medical institutions. Nurses, who have been considered the backbone of the health care system, are prone to LBP given the work-related stressors they experience. Quite a large number of studies have shown that work-related injuries are quite common among members of the allied health care team. Moreover, studies revealed that low backache has been ranked third among the conditions nursing professionals experience [7,11,14,16]. Our results have been consistent with those presented in another study recently conducted in Zimbabwe that evaluated and reported the effects of various work-related factors on work-related musculoskeletal disorders among registered nurses in Hospitals across Harare [15]. The current study found that 74.8% of nursing professionals suffered from LBP, which was similar to that reported by other studies in other parts of the world. In a Nigerian study, Tinubu et al. reported a LBP prevalence of 78% among Nigerian nurses [16]. Yet another study conducted among Chinese nurses revealed a LBP prevalence of 77.4% among Chinese nurses [20]. Moreover, a series of studies in a large university hospital in Switzerland reported a high prevalence of LBP (73–76%) among nurses [21].

Among the various work-related factors investigated herein, such as education level, years of experience, place of work, nature of work, and direct contact hours per week with patients, only the last three were found to be significantly associated with LBP (*p* < 0.05). Though many of the symptoms and etiology of LBP remain unclear, investigations have linked work-related factors to LBP onset [12,18]. The manual tasks that most nurses perform daily have been shown to contribute toward LBP [19,22,23]. Stressful trunk postures that frequently occur due to direct patient contact during prolonged working conditions can provoke low back symptoms. The results presented herein clearly showed that the nature of work (i.e., 80% of the nurses who participated in this study in Najran with LBP reported working full time, while 74.2% reported having more than 30–40 h of direct patient contact) was a major work-related factor associated with LBP among nurses.

Nurses have increased susceptibility for developing back pain due to various stressors associated with long working hours. A series of studies found that LBP was positively related to the number of working hours [13,14,24,25].

The current study reported that LBP was more prevalent among female (60.8%) than male (39.2%) nurses. Another study conducted in Nigeria observed that among the 148 and 260 male and female nurses analyzed, 96 and 204 reported to have LBP, respectively. Our results also showed that that majority of nurses (76.2%) who developed some type of LBP were between the ages of 26 to 40 years, with only 7.5% and 1.8% of those aging 41–50 and >50 years developing LBP, respectively. However, no significant association has been observed between age and LBP (Table 2). This finding is similar to that presented in another set of studies conducted in Nigeria wherein age was not correlated with work-related musculoskeletal disorders among nursing professionals [16]. However, some studies have reported that the prevalence of LBP increases with age. Accordingly, Kassebaum et al. previously reported that susceptibility to the development of chronic diseases typically increases with age. This phenomenon is a reflection of genetic risk factor exposure and physiological changes along with the cumulative effect of environmental stress (occupation) [7]. An interesting finding observed herein was that a majority (76.2%) of nurses who reported having LBP were in the 26 to 40 year age group, which can be considered quite young by any standards. Thus, the reported prevalence of LBP among the relatively young population may be assumed to be caused by work-related stresses rather than actual chronic diseases or age-related changes, which could enhance the severity and prevalence.

Apart from nature of work, the current study found that place of work, working hours, poor working habits, and incorrect lifting postures can be associated with LBP. Another point emphasized herein is that more female nurses (60.8%) reported having LBP compared to male nurses (39.2%). This could perhaps be explained by the higher amount of work pressure in obstetrics and gynecology units, including labor wards, which are major parts of general hospitals, compared to that in other areas, such as university teaching hospitals or rehabilitation centers.

Our results found that 89.3% of nurses reported no past medical conditions. Considering that the majority of the nurses were relatively young (26–40 years), no significant association has been observed (Table 3). Our findings also determined that 59.3% of the nurses reported having acute LBP, while almost 84% reported having experienced LBP for only one to four weeks. This could have been caused by working without changing positions for a prolonged period, as well as lifting/transferring patients and increased patient load, which have been previously linked to acute LBP which can be reversed by taking rests.

Another important finding reported herein is that 88.2% of nursing professionals had developed mild to moderate LBP, with 66.4% reporting no functional limitations. This may have been due to the fact that 90% of the nurses had localized LBP without radiation. Our findings were similar to those presented in a study conducted on physical therapists in Kuwait, which reported that the phenomenon of peripheralization could adversely affect functional movements [26]. Lastly, the present study found that 27.2% of nurses experienced difficulty in lifting and carrying weight, 24.2% experienced difficulty with prolonged standing, 20.3% experienced difficulty in handling patients, and 18.6% reported having problems pulling and pushing weights (Table 3). This might be associated with decreasing nurses’ work capacity, which might lead to sick leave, and might encourage managers to do something about the situation. The aforementioned difficulties are inherent in the various work stressors associated with the nursing profession. Accordingly, a number of previous studies have reported that LBP was strongly associated with working in the same position for prolonged periods, lifting/transferring of patients, and increased patient load. Our findings are consistent with those of the Zimbabwean study, which found that repeated performance of various work-related tasks, caring for a large number of patients daily, awkward bending or twisting of the back, lifting or transferring dependent patients, and heavy equipment and materials were significantly associated with the incidence of LBP [15].

There are several limitations in our study, one of them regarding selection bias. We know little about which individuals wanted to answer the questionnaires: maybe those with LBP were more interested in answering the study than those without back pain. Another limitation is the relatively small sample size of the participants in this study.

## 5. Conclusions

The present study concluded that nursing professionals working across various institutions in the Najran region of Saudi Arabia had a high prevalence of LBP. Moreover, our findings showed that various work-related characteristics, including place and nature of work, might be associated with the onset of LBP. The result presented herein will be of considerable help to institutional heads in their establishment of related policies and appropriate measures in managing associated work-related LBP among not only nursing professionals but also other related health care professionals.

## Figures and Tables

**Table 1 medsci-08-00045-t001:** Participant demographics.

Characteristics	Level	Number out of Total 187	Percentage %
Nationality	Saudi	66	35.3
	Non-Saudi	121	64.7
Age (in years)	21–30	83	44.4
	31–40	83	44.4
	41–50	16	8.5
	>50	5	2.7
Gender	Male	79	42.2
	Female	108	57.8
Educational qualification	Diploma	58	31
	Bachelor’s degree	113	60.4
	Master’s degree and higher	16	8.6
Work experience	<10	110	58.8
	10–19	58	31
	>19	19	10.2
Place of work	General hospital	111	59.4
	Private	65	34.7
	Others	11	5.9
Total		187	100

**Table 2 medsci-08-00045-t002:** Sociodemographic and work-related characteristics and their association with low back pain.

Characteristics	Level	Nurses with LBP (*n* = 140)	Nurses without LBP (*n* = 47)	*p* Value
Nationality	Saudi	45 (68.2%)	21 (31.8%)	0.120
	Non-Saudi	95 (78.5%)	26 (21.5%)	
Age	21–25	19 (67.9%)	9 (32.1%)	0.364
	26–30	41 (74.5%)	14 (25.5%)	
	31–35	41 (82%)	9 (18%)	
	36–40	25 (75.8%)	8 (24.2%)	
	41–45	9 (69.2%)	4 (30.8%)	
	46–50	3 (100%)	0 (0%)	
	>50	2 (40%)	3 (60%)	
Sex	Male	55(69.62 %)	24 (30.38%)	0.157
	Female	85(78.7%)	23 (21.3%)	
Education level	Diploma	46 (79.3%)	12 (20.7%)	0.382
	Bachelor’s degree	84 (74.3%)	29 (25.7%)	
	Master’s degree and higher	10 (62.5%)	6 (37.5%)	
Work experience	<5 years	120 (76.4%)	37 (23.6%)	0.292
	5–9 years	1 (33.3%)	2 (66.7%)	
	10–14 years	1 (100%)	0 (0%)	
	>15 years	18 (69.2%)	8 (30.8%)	
Workplace	General hospital	88 (79.3%)	23 (20.7%)	0.196
	Private Clinic/Hospital	14 (63.6%)	8 (36.4%)	
	University clinics/hospitals	3 (50%)	3 (50%)	
	Others including rehabilitation centers	35 (72.9%)	13 (27.1%)	
Nature of work	Full time	112 (78.9%)	30 (21.1%)	0.025 *
	Part time	28 (62.2%)	17 (37.8%)	
Hours with patients/week	<10	7 (87.5%)	1 (12.5%)	0.870
	10–19	18 (78.3%)	5 (21.7%)	
	20–29	11 (78.6%)	3 (21.43%)	
	30–39	50 (74.6%)	17 (25.4%)	
	>40	54 (72%)	21 (28%)	

* *p* < 0.05.

**Table 3 medsci-08-00045-t003:** Characteristics of low back pain among the included participants.

Characteristics	Characteristics	Number of Participants with LBP (*n* = 140)	Frequency (%)
Past medical condition:	None	125	89.3%
	Cardiac conditions	5	3.6%
	Hypertension	4	2.9%
	Others	6	4.3%
Nature of lower back pain (LBP):	Acute	83	59.3%
	Chronic	57	40.7%
Average duration of LBP:	1 week	79	58.4%
	2–4 weeks	35	25%
	>4 weeks	26	18.6%
Severity of LBP:	Mild	63	45%
	Moderate	52	37.2%
	Severe	25	17.8%
Functional limitation resulting from LBP:	None	93	66.4%
	Mild	33	23.6%
	Moderate	10	7.1%
	Severe	4	2.9%
Effect of LBP on normal activities:	Yes	103	73.6%
	No	37	26.4 %
Anatomical location of pain on a body chart:	Lower back	72	51.4%
	Lower back and buttocks	54	38.6%
	Lower back and thighs	5	3.5%
	Lower back and leg	4	3%
	Lower back and both legs	5	3.5%
Regular activities affected by LBP: *n* = 103	Lifting, carrying	28	27.2%
	Patient handling	21	20.3%
	Prolonged sitting	10	9.7%
	Prolonged standing	25	24.2%
	Pulling, pushing	19	18.6%

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
