# Peer review of "Prevalence of Low Back Pain among Nursing Staff in Najran, Saudi Arabia: A Cross-Sectional Study"

_medsci, 2020, doi:10.3390/medsci8040045_

Round 1

Reviewer 1 Report

The author can be congratulated with this interesting cross-sectional study investigating the prevalence of low back pain among nursing staff in a region in Saudi Arabia. The results show that the prevalence of low back pain is common among nurses, which has also been shown in previous studies. The study also shows other interesting characteristics of the back pain. The paper is mostly well-written and the English clear, but some parts need to be rephrased. The discussion needs more attention.

Detailed comments:

  1. Subjects and methods

I suggest it would be better to have the heading “2. Methods and subjects” or even better just “2. Methods”.

Under methods there is missing information about the content of the questions about prevalence etc. Where do the questions come from? Previous studies? Validity and reliability of the questions and questionnaires used? References? Please provide more details.

Did you use electronic or paper questionnaires or both?

What was the ethics approval number?

  1. Results

When did you collect the data? Years? Months?

Page 2 line 83

The following text should be placed under “Methods” (not under Results).

“Data were initially coded and keyed into a Microsoft Excel spreadsheet and then transferred to SPSS software for data processing. Descriptive analysis was used to summarize categorical data. The Pearson chi-square test was used as part of inferential statistics to determine the degree of association between categorical variables. A p value of <0.05 indicated significant differences between variables.”

Table 2

Please explain the abbreviations LBA and LBP. Why do you use both the terms LBA and LBP?

According to my Pearson chi-square test the difference between genders is not significant. Please reconsider the p-value of 0.042.

Please explain what you mean by “*”.

To investigate differences in age, it might be more appropriate to use the Mann-Whitney U test? Have you done that? And is it still not significant?

Page 4 line 117

"Overall, our study observed that 62.8% of the nurses working at general hospitals developed varying degrees of LBP, whereas only 14% and 21% of those working at university hospitals and private clinics or hospitals developed LBP, respectively." This way of writing may lead to misinterpretation of the results and should be rephrased. För example, in total 101 nurses worked at general hospitals, and of these, 79.3% developed varying degrees of LBP. Please check all other text where you use percentages as well. it is very easy to mix vertical and horizontal percentage figures.

Table 3.

Please report the total number of nurses with LBP.

Regarding "Average duration of LBP" it is more appropriate to place "Chronic" below ">4 weeks".

Discussion

page 5 line 162

"LWP prevalence" should be "LBP prevalence"?

page 5 line 164.

I am missing a critique of the study regarding selection bias. We know very little about which individuals that wanted to answer the questionnaires: maybe those with LBP were more interested in answering the study than those without back pain? Maybe there is a healthy worker effect? Please report on that under discussion.

page 5 line 173

"80% of the nurses in Najran reported working full time" should be 80% of the nurses in Najran with LBP reported working full time"?

page 6 line 181

"The current study reported that LBP was significant more prevalent among female (60.8%) than 182 male (39.2%) nurses." As I pointed out previously there was no siginificant difference. Please rephrase the text. Or have I made a  mistake in my calculations?

pager 6 line 199

"Apart from nature of work, the current study found that place of work, working hours, poor 200 working habits, and incorrect lifting postures could be important causes for LBA." Considering the design of the study, you cannot claim that you investigated "causes" for LBP. You may have found an association, but that is not the same. Please rephrase.

page 6 line 201

"..nurses working at general hospitals reported the 202 highest incidence of LBA (62.8%)." As mentioned previously, this figure is not correct. Please correct this figure.

page 6 line 220.

"...27.2% of the nurses experienced difficulty in lifting and carrying weight, 24.2% experienced difficulty with prolonged standing, 20.3% experienced difficulty in handling patients, and 18.6% reported having problems pulling and pushing weight". Here you could mention that this might decrease the nurses work capacity, which might lead to sick leave, and might encourage managers to do something about the situation.

There is a lack of description of limitations of the study. Please add this to the Discussion.

Conclusion

Page 7 line 234

“..various work-related characteristics, including place and nature of work, play a significant role in the onset of LBP.” You cannot claim this because you have not investigated causal relations. You can write that you found associations. Please rephrase.

Author Response

Thank you for your constructive feedback.

Reviewer comments

Author Response.

  1. Subjects and methods

I suggest it would be better to have the heading “2. Methods and subjects” or even better just “2. Methods”.

Under methods there is missing information about the content of the questions about prevalence etc. Where do the questions come from? Previous studies? Validity and reliability of the questions and questionnaires used? References? Please provide more details.

Did you use electronic or paper questionnaires or both?

What was the ethics approval number?

  1. Results

When did you collect the data? Years? Months?

Page 2 line 83

The following text should be placed under “Methods” (not under Results).

“Data were initially coded and keyed into a Microsoft Excel spreadsheet and then transferred to SPSS software for data processing. Descriptive analysis was used to summarize categorical data. The Pearson chi-square test was used as part of inferential statistics to determine the degree of association between categorical variables. A p value of <0.05 indicated significant differences between variables.”

Table 2

Please explain the abbreviations LBA and LBP. Why do you use both the terms LBA and LBP?

According to my Pearson chi-square test the difference between genders is not significant. Please reconsider the p-value of 0.042.

Please explain what you mean by “*”.

To investigate differences in age, it might be more appropriate to use the Mann-Whitney U test? Have you done that? And is it still not significant?

Page 4 line 117

"Overall, our study observed that 62.8% of the nurses working at general hospitals developed varying degrees of LBP, whereas only 14% and 21% of those working at university hospitals and private clinics or hospitals developed LBP, respectively." This way of writing may lead to misinterpretation of the results and should be rephrased. För example, in total 101 nurses worked at general hospitals, and of these, 79.3% developed varying degrees of LBP. Please check all other text where you use percentages as well. it is very easy to mix vertical and horizontal percentage figures.

Table 3.

Please report the total number of nurses with LBP.

Regarding "Average duration of LBP" it is more appropriate to place "Chronic" below ">4 weeks".

Discussion

page 5 line 162

"LWP prevalence" should be "LBP prevalence"?

page 5 line 164.

I am missing a critique of the study regarding selection bias. We know very little about which individuals that wanted to answer the questionnaires: maybe those with LBP were more interested in answering the study than those without back pain? Maybe there is a healthy worker effect? Please report on that under discussion.

page 5 line 173

"80% of the nurses in Najran reported working full time" should be 80% of the nurses in Najran with LBP reported working full time"?

page 6 line 181

"The current study reported that LBP was significant more prevalent among female (60.8%) than 182 male (39.2%) nurses." As I pointed out previously there was no siginificant difference. Please rephrase the text. Or have I made a  mistake in my calculations?

pager 6 line 199

"Apart from nature of work, the current study found that place of work, working hours, poor 200 working habits, and incorrect lifting postures could be important causes for LBA." Considering the design of the study, you cannot claim that you investigated "causes" for LBP. You may have found an association, but that is not the same. Please rephrase.

page 6 line 201

"..nurses working at general hospitals reported the 202 highest incidence of LBA (62.8%)." As mentioned previously, this figure is not correct. Please correct this figure.

page 6 line 220.

"...27.2% of the nurses experienced difficulty in lifting and carrying weight, 24.2% experienced difficulty with prolonged standing, 20.3% experienced difficulty in handling patients, and 18.6% reported having problems pulling and pushing weight". Here you could mention that this might decrease the nurses work capacity, which might lead to sick leave, and might encourage managers to do something about the situation.

There is a lack of description of limitations of the study. Please add this to the Discussion.

Conclusion

Page 7 line 234

“..various work-related characteristics, including place and nature of work, play a significant role in the onset of LBP.” You cannot claim this because you have not investigated causal relations. You can write that you found associations. Please rephrase.

Done.

Edited in the text.

Electronic, edited in the text.

Added in the text.

Added in the text.

Moved.

Changed to LBP throughout the manuscript.

  *P<0.05

I used Pearson chi-square

Nice catch! I edited it to “who participated in our study.”

I did.

OK.

Edited.

Edited.

Edited.

Changed.

I edited “who participated in our study.”

Edited.

Added.

Changed.

Reviewer 2 Report

This is a review of the manuscript “Prevalence of low back pain among nursing staff in Najran, Saudi Arabia: A cross-sectional study.  The manuscript presents a straight forward study of nursing staff and the prevalence of low back pain.  A few specific comments below.

  1. The authors start with equating low back pain (LBP) and low back ache (LBA). The switching between LBP and LBA just seems like it will muddy an already complex issue of low back pain.  Most of the literature uses low back pain therefore I would like to see the manuscript revised to only use LBP instead of using LBA.   
  2. Recruitment of subjects. Did the authors really electronically send out 187 questionnaires to nursing professionals and received responses from all 187?  How many nurses were approached to participate?  What was the participation percentage?
  3. Line 66 states that the questionnaire was modified. How was the questionnaire modified?  Could the authors add the questionnaire to the manuscript as an appendix?
  4. In Table 1. The first row is Character, level, Number out of total 62, Percentages. The number out of 62 does not make sense or is not clear?
  5. Line 97, states while approximately 91% of the participants had a bachelor’s degree in nursing. This does not agree with the Table 1, which shows 113 participants 60.4% had bachelor’s degree.  What is the “diploma” in Table 1. 
  6. Table 2 heading PLEASE change nurses with Low back ache to nurses with LBP.
  7. Table 2, What is the * by the p-value suppose to indicate?
  8. Line 113, nearly all 92.8% had bachelor’s degrees. Here the author appears to combine diploma and bachelor’s degree from table 1 and 2.  Is there really a difference between these or should these be combined everywhere?
  9. Line 138. Our rests showed that majority of the …. What is “our rests”?
  10. Nice comparison of the results of the current study to the literature in the first paragraph.
  11. Discussion, Line 200. Please use LBP and not LBA.  The use of LBA will just be confusing to readers. 

Author Response

Thank you for your constructive feedback regarding this article:

1. The authors start with equating low back pain (LBP) and low back ache (LBA). The switching between LBP and LBA just seems like it will muddy an already complex issue of low back pain.  Most of the literature uses low back pain therefore I would like to see the manuscript revised to only use LBP instead of using LBA. The authors introduce LBA and then use it in Table 2  

1. Edited throughout the article, Thank you.

2. Recruitment of subjects. Did the authors really electronically send out 187 questionnaires to nursing professionals and received responses from all 187?  How many nurses were approached to participate?  What was the participation percentage?

2. It is edited in the manuscript. It was electronically and was sent through
various social media sites and 187 responses were received.

3. Line 66 states that the questionnaire was modified. How was the questionnaire modified?  Could the authors add the questionnaire to the manuscript as an appendix?

3. I already added it to my files. I will make sure you read it.

4. In Table 1. The first row is Character, level, Number out of total 62, Percentages. The number out of 62 does not make sense or is not clear?

4. Good catch! Edited.

5. Line 97, states while approximately 91% of the participants had a bachelor’s degree in nursing. This does not agree with the Table 1, which shows 113 participants 60.4% had bachelor’s degree.  What is the “diploma” in Table 1. 

5. Edited, Thanks!

6. Table 2 heading PLEASE change nurses with Low back ache to nurses with LBP.

6. Changed.

7. Table 2, What is the * by the p-value suppose to indicate?

7. Edited in the table

8. Line 113, nearly all 92.8% had bachelor’s degrees. Here the author appears to
combine diploma and bachelor’s degree from table 1 and 2. Is there really a
difference between these or should these be combined everywhere?

8. Edited.

9. Line 138. Our rests showed that majority of the …. What is “our rests”?

9. Edited.

10. Discussion. Nice comparison of the results of the current study to the literature in the first paragraph.

10. Thank you!

11. Discussion, Line 200. Please use LBP and not LBA. The use of LBA will just be
confusing to readers.

11. Edited.

Round 2

Reviewer 1 Report

The author can be congratulated with this interesting cross-sectional study investigating the prevalence of low back pain among nursing staff in a region in Saudi Arabia.

The author has made several valuable revisions. A major concern is still some statistical calculations and interpretation of the results. In that critical area nothing has been changed. According to my knowledge some calculations do not show correct figures and some calculations should be performed using other statistical methods. As a result, interpretations are provided that are not correct.

Detailed comments:

  1. Subjects and methods

“Dionee et al.” should be “Dionne et al.”

Under methods there is missing information about the questions. Please add that you did your own translation of the items and into which language.

  1. Results

Table 2

According to my Pearson chi-square test the difference between genders is not significant. Please reconsider the p-value of 0.042.

To investigate differences in age, it might be more appropriate to use the Mann-Whitney U test? Have you done that? And is it still not significant?

Page 4 line 116

"Overall, our study observed that 62.8% of the nurses working at general hospitals developed varying degrees of LBP, whereas only 14% and 21% of those working at university hospitals and private clinics or hospitals developed LBP, respectively." This way of writing may lead to misinterpretation of the results and should be rephrased. For example, in total 101 nurses worked at general hospitals, and of these, 79.3% developed varying degrees of LBP. Please check all other text where you use percentages as well. it is very easy to mix vertical and horizontal percentage figures.

Table 3.

Please report the total number of nurses with LBP.

Discussion

page 6 line 184

"The current study reported that LBP was significant more prevalent among female (60.8%) than 182 male (39.2%) nurses." As I pointed out previously there was no significant difference. Please rephrase the text. Or have I made a  mistake in my calculations?

pager 6 line 202

"Apart from nature of work, the current study found that place of work, working hours, poor 200 working habits, and incorrect lifting postures could be important causes for LBA." Considering the design of the study, you cannot claim that you investigated "causes" for LBP. You may have found an association, but that is not the same. Please rephrase.

page 6 line 204

"..nurses working at general hospitals reported the highest incidence of LBP (62.8%)." As mentioned previously, this figure is not correct. Please correct this figure.

page 6 line 226.

“that this might decrease the nurses work capacity, which might lead to sick leave, and might encourage managers to do something about the situation.” Please rephrase the English sentence.

Author Response

Thank you for your excellent feedback.

The author has made several valuable revisions. A major concern is still some statistical calculations and interpretation of the results. In that critical area nothing has been changed. According to my knowledge some calculations do not show correct figures and some calculations should be performed using other statistical methods. As a result, interpretations are provided that are not correct.

Done.

Detailed comments:

Subjects and methods

“Dionee et al.” should be “Dionne et al.”

Done.

Under methods there is missing information about the questions. Please add that you did your own translation of the items and into which language.

The questionnaire was in English and was not translated. Since health care provider should have a good English language background. 

Results

Table 2

According to my Pearson chi-square test the difference between genders is not significant. Please reconsider the p-value of 0.042.

Edited.

To investigate differences in age, it might be more appropriate to use the Mann-Whitney U test? Have you done that? And is it still not significant?

We did not consider the pain grades for the participants, we just looked whether they have low back pain or not.

Page 4 line 116

"Overall, our study observed that 62.8% of the nurses working at general hospitals developed varying degrees of LBP, whereas only 14% and 21% of those working at university hospitals and private clinics or hospitals developed LBP, respectively." This way of writing may lead to misinterpretation of the results and should be rephrased. For example, in total 101 nurses worked at general hospitals, and of these, 79.3% developed varying degrees of LBP. Please check all other text where you use percentages as well. it is very easy to mix vertical and horizontal percentage figures.

Nice observation, edited.

Table 3.

Please report the total number of nurses with LBP.

Edited.

Discussion

page 6 line 184

"The current study reported that LBP was significant more prevalent among female (60.8%) than 182 male (39.2%) nurses." As I pointed out previously there was no significant difference. Please rephrase the text. Or have I made a mistake in my calculations?

Nice catch! I edited it.

pager 6 line 202

"Apart from nature of work, the current study found that place of work, working hours, poor 200 working habits, and incorrect lifting postures could be important causes for LBA." Considering the design of the study, you cannot claim that you investigated "causes" for LBP. You may have found an association, but that is not the same. Please rephrase.

Done.

page 6 line 204

"..nurses working at general hospitals reported the highest incidence of LBP (62.8%)." As mentioned previously, this figure is not correct. Please correct this figure.

Removed.

page 6 line 226.

“that this might decrease the nurses work capacity, which might lead to sick leave, and might encourage managers to do something about the situation.” Please rephrase the English sentence.

Edited.

Reviewer 2 Report

The changes made to the manuscript are adequate to move forward with publication.  Nice work very quick.

Author Response

Thank you

This manuscript is a resubmission of an earlier submission. The following is a list of the peer review reports and author responses from that submission.